# New Titanium(IV)-Alkoxide Complexes Bearing Bidentate OO Ligand with the Camphyl Linker as Catalysts for High-Temperature Ethylene Polymerization and Ethylene/1-Octene Copolymerization

**DOI:** 10.3390/polym14214735

**Published:** 2022-11-04

**Authors:** Vladislav A. Tuskaev, Svetlana Ch. Gagieva, Dmitrii A. Kurmaev, Yulia V. Nelyubina, Petr V. Primakov, Maria D. Evseeva, Evgenii K. Golubev, Mikhail I. Buzin, Galina G. Nikiforova, Pavel B. Dzhevakov, Viktor I. Privalov, Kasim F. Magomedov, Boris M. Bulychev

**Affiliations:** 1Department of Chemistry, M. V. Lomonosov Moscow State University, 1 Leninskie Gory, 119992 Moscow, Russia; 2A. N. Nesmeyanov Institute of Organoelement Compounds, Russian Academy of Sciences, Vavilova St. 28, 119991 Moscow, Russia; 3Department of Chemistry, Mendeleev University of Chemical Technology of Russia, Miusskaya pl. 9, 125047 Moscow, Russia; 4Enikolopov Institute of Synthetic Polymer Materials, Russian Academy of Sciences, Profsoyuznaya Str. 70, 117393 Moscow, Russia; 5Kurnakov Institute of General and Inorganic Chemistry, Russian Academy of Sciences, 31 Leninsky Prospect, 119991 Moscow, Russia

**Keywords:** Ti(IV) complexes, OO-ligand, thermal stability, ultra-high molecular weight polyethylene, polyolefin elastomers

## Abstract

In order to increase the thermal stability of olefin polymerization precatalysts, new titanium(IV) complexes with diolate ligands differing in the degree of steric hindrances were synthesized from readily available precursor (±)camphor. The structures of the complexes **1–2** were established by X-ray diffraction. Complexes **1–4** in the presence of an activator {Et_n_AlCl_3-n_ + Bu_2_Mg} catalyzed the synthesis of UHMWPE with an M_v_ up to 10 million and a productivity of up to 3300 kg/mol_Ti_·atm·h. The obtained polymers are obviously characterized by a low density of macromolecular entanglement, which makes it possible to use the solid-phase method for their processing. The mechanical characteristics of the oriented UHMWPE films had a breaking strength up to 2.7 GPa and an elastic modulus of up to 151 GPa. The precatalysts **1–4** were also active in ethylene/1-octene copolymerization. The comonomer content was in the range of 1.4–4.6 mol%. The use of a rigid linker and an increase in the steric load of the diolate complexes ensured the thermal stability of the catalytic system in the range of 50–70 °C.

## 1. Introduction

The development of catalysts for the (co)polymerization of olefins based on transition metal complexes is one of the most advanced areas of modern organometallic chemistry [1,2,3,4]. However, research in this field will undoubtedly continue due to the constant need for new polymer materials. The value of new catalytic systems is determined not only by their productivity and the properties of the resulting polymers, but also by the technological parameters of the polymerization process, including the ability to operate in an acceptable temperature range.

Titanium-alkoxide complexes are perhaps one of the most accessible and inexpensive precatalysts. We know that titanium alkoxides in the presence of organoaluminum compounds are capable of catalyzing the polymerization of conjugated dienes [5,6,7] as well as the oligomerization of ethylene. The best-known example is the alphabutol process, wherein the combination of Ti(OR)_4_–AlEt_3_ is employed for the highly selective dimerization of ethylene to 1-butene [8].

At the same time, simple titanium alkoxides and more sophisticated alkoxo–titanium complexes have rarely been used for the polymerization of olefins. The ability to polymerize ethylene in the presence of activators traditional for the Ziegler–Natta catalysis is possessed by titanium–alkoxo complexes containing additional chlorine atoms or other donor atoms. Thus, the dichlorotitanium-alkoxide complex [(HOEt)Ti(μ-OEt)OEt(Cl)_2_]_2_ (Compound **I**, [Fig polymers-14-04735-ch001]), activated with MAO, catalyzes ethylene polymerization with an activity of up to 750 kg/mol h and propylene (up to 87 kg/mol h). This precatalyst unexpectedly displays a single-site behavior [9]. The alkoxo complexes **II–III** activated with MAO or AlEt_3_ catalyze the formation of polyethylene along with a percentage of oligomers. The bischelated complex **III** appears to be the most active [10]. A titanium(IV)–dimeric complex **IV** stabilized by a benzoin derivative in the presence of MAO catalyzes ethylene polymerization with an activity up to 300 kg/mol·atm·h as well as the copolymerization of ethylene with norbornene The catalytic systems are characterized by a long lifetime and the ability to produce high molecular weight linear PE and vinyl-type PNB [11]. One of the few examples of homoleptic and heteroleptic Ti(IV)–alkoxo complexes **V–VI** capable of polymerizing ethylene without the formation of oligomeric products is given in [12]. In the presence of Et_3_Al_2_Cl_3_, these compounds catalyzed the formation of low molecular weight PE.

As was shown by Y. V. Kissin et al., the addition of the organomagnesium compound, e.g., Bu_2_Mg to the mixture Ti(OiPr)_4_–Et_2_AlCl results in active, cheap, and affordable catalytic systems suitable for the polymerization of propylene [13] and the copolymerization of ethylene with higher olefins [14].

Previously, we reported on the ability of various titanium(IV)–diolate complexes **VII-X** [15,16,17,18,19,20,21,22], including those with additional heteroatoms [23,24,25,26,27] ([Fig polymers-14-04735-ch001]), to catalyze the (co)polymerization of ethylene in the presence of such Al/Mg activators. This group of postmetallocene precatalysts is poorly studied, which is most likely due to the specifics of their activation: in the presence of trialkylaluminum derivatives, alkylaluminum chlorides, or alkylalumoxanes (traditional activators in Ziegler–Natta catalysis), and the activity of these systems is low or does not manifest itself at all. However, when using the Al/Mg activators Alk_n_AlCl_3-n_+Bu_2_Mg (proposed by Yu. V. Kissin et al. [28,29]), their productivity in ethylene polymerization reached 4000 kg/mol·atm·h. It is important to note that, in most cases, these catalytic systems produce disentangled UHMWPE, which can be processed by the solid-phase method into high strength-oriented films and tapes, which are in high demand in various industries. In addition, such catalytic systems effectively catalyze the copolymerization of ethylene with higher olefins, and in some cases, even higher productivity is achieved ofup to 5 tons of copolymer /mol atm·h [20].

An essential point limiting the possibility of the industrial implementation of this group of catalysts is their low thermal stability. As a rule, the maximum productivity of such systems is achieved at a temperature of 30 °C [15,16,17,18,19,20,21,22,23,24,25,26,27]; an increase in the polymerization temperature to 50–80 °C is accompanied by deactivation and a significant reduction in the polymer’s molecular weight. The aim of this work is the structural modification of the ligand environment of the metal, aimed at increasing the thermal stability of the considered precatalysts.

## 2. Experimental Section

All manipulations with air-sensitive materials were performed using standard Schlenk techniques. Argon and ethylene of a special purity grade (Linde gas) were dried by purging through Super Clean™ Gas Filters.

Toluene and nefras were distilled over Na/benzophenone ketyl, and the water content was periodically controlled by Karl Fischer coulometry by using a Methrom 756 KF apparatus. Diethylaluminum chloride, ethylaluminum sesquichloride, and di-*n*-butylmagnesium (Aldrich) were used without further purification. (±)-Camphorquinone was obtained by the method described in [30]. The preparation of the ligands **L1** and **L2** followed the procedure described in [31]; their properties corresponded to the literature data.

NMR spectra were recorded on a Bruker AMX-400 instrument (Mundelein, Illinois 60060 USA). Elemental analysis (C, H, Cl) was performed by the microanalytical laboratory at A. N. Nesmeyanov Institute of Organoelement Compounds on Carlo Erba-1106 and Carlo Erba-1108 instruments. The content of Ti was performed by X-ray fluorescence analysis on a VRA-30 device (Karl Zeiss, Germany).

**[L^1^Ti(OiPr)_2_]_2_** (Complex **1**). Ligand **L1** (0.85 g, 5 mmol) and toluene (22 mL) were placed into a Schlenk tube equipped with a magnetic stirrer under an argon atmosphere, followed by the addition of Ti(OiPr)_4_ (1.42 g, 1.48 mL, and 5 mmol) at room temperature. The resulting suspension was heated until all solids dissolved. The next day, the formed crystals were collected by filtration and dried in vacuo. Yield 1.41 g (81.5%). Calculated (%) for C_32_H_60_O_8_Ti_2_ (668.55): C, 57.5; H, 9.0; O, 19.1; and Ti, 14.3. Found (%): C, 57.2; H, 8.6; and Ti, 14.0. ^1^H NMR (400 MHz, CDCl_3_), δ: 0.82 (s, 3H), 0.85 (s, 3H), 0.98 (s, 3H), 1.07 (s, 6H), 1.18 (s, 6H), 1.44 (s, 2H), 1.67 (s, 1H), 1.96 (s, 2H), 4.03 (d, J = 20.2 Hz, 2H), and 4.32 (d, J = 23.0 Hz, 2H). ^13^C NMR (101 MHz, CDCl_3_), δ: 88.18, 84.05, 77.35, 77.03, 76.71, 67.81, 64.45, 48.91, 47.87, 46.31, 31.90, 25.36, 24.30, 24.27, 23.65, 23.39, 23.25, 20.78, 20.41, and 10.67.

**[L^2^Ti(OiPr)_2_]_2_** (Complex **2**) was obtained by a similar method. Yield 1.34 g (74.2%). Calculated (%) for C_36_H_68_O_8_Ti_2_ (724.65): C, 59.7; H, 9.5; O, 17.7; and Ti, 13.2. Found (%): C, 59.4; H, 9.2; and Ti, 13.0. ^1^H NMR (400 MHz, CDCl_3_), δ: 0.87 (s, 3H), 0.89 (s, 3H), 0.96 (s, 3H), 1.09 (s, 6H), 1.12 (s, 3H), 1.20 (s, 3H), 1.23 (s, 3H), 1.65 (s, 3H), 1.94 (dd, J = 29.4, 4.9 Hz, 2H), 3.70 (s, 1H), 4.09 (m, 2H), and 4.36 (m, 2H). ^13^C NMR (101 MHz, CDCl_3_), δ: 80.39, 80.58, 77.23, 56.85, 53.20, 48.38, 30.99, 26.06, 25.62, 25.36, 24.36, 23.36, 23.23, 22.89, 21.53, 10.70, and 10.58.

**L^1^TiCl_2_ 2iPrOH** (Complex **3**). In a 100 mL flame-dried Schlenk flask, ligand **L1** (0.85 g, 5 mmol) was dissolved in anhydrous toluene (20 mL). A solution of TiCl_2_(OiPr)_2_ (1.185 g, 0.05 mmol) in toluene (20 mL) was added to the resulting solution under stirring in an argon atmosphere. The solution was stirred for 14 h at room temperature, and the precipitated complex was filtered off, washed with hexane (5 mL), and dried in a vacuum. The yield was 1.4 g (69%). Calculated (%) for C_16_H_32_Cl_2_O_4_Ti (407.19): C, 47.2; H, 7.9; Cl, 17.4; O, 15.7; and Ti, 11.8. Found (%): C, 46.6; H, 7.6; Cl, 17.2; and Ti, 11.5. ^1^H NMR (400 MHz, CDCl_3_), δ: 0.88 (s, 3H), 0.92 (dd, J = 38.4, 11.9 Hz, 6H), 1.01 (s, 6H), 1.05 (s, 3H), 1.15 (s, 3H), 1.51 (s, 2H), 1.73 (d, J = 4.8 Hz, 1H), 2.04 (d, J = 5.0 Hz, 2H), 4.20 (s, 2H), and 4.39 (s, 2H). ^13^C NMR (101 MHz, CDCl_3_), δ: 87.99, 83.87, 67.62, 64.26, 48.72, 47.68,46.12, 31.71, 25.17, 24.12, 24.09, 23.47, 23.06, 20.59, 20.22, and 10.48.

**L^2^TiCl_2_ 2iPrOH** (Complex **4**) was obtained by a similar method. Calculated (%) for C_18_H_36_Cl_2_O_4_Ti (435.25): C, 49.7; H, 8.3; Cl, 16.3; O, 14.7; and Ti, 11.0. Found (%): C, 49.3; H, 8.0; Cl, 16.1; and Ti, 10.6. ^1^H NMR (400 MHz, CDCl_3_), δ: 0.89 (s, 3H), 0.92 (s, 3H), 0.96 (s, 6H), 1.20 (s, 6H), 1.32 (s, 3H), 0.89 (s, 3H), 1.92 (s, 3H), 1.68 (m, 2H, CH_2_), 2.19 (m, 2H), 2.42 (s, 2H), 4.06 (m, 2H), and 4.78 (m, 1H).

### 2.1. X-ray Crystal Structure Determination

X-ray diffraction experiments were carried out at 100 K for **1** and at 240 K for **2** (below this temperature, the crystals of **2** cracked) with a Bruker D8 Quest diffractometer, using graphite monochromated Mo-Kα radiation (λ = 0.71073 Å). Using Olex2 [32], the structures were solved with the ShelXT [33] structure solution program using Intrinsic Phasing and refined with the olex2.refine [34] refinement package using Least-Squares minimization against F^2^ in anisotropic approximation for nonhydrogen atoms. Positions of hydrogen atoms were calculated, and they were refined in isotropic approximation within the riding model. Crystal data and structure refinement parameters for **1** and **2** are given in Table 1. CCDC 2189447 (for **1**) and 2189448 (for **2**) contain the supplementary crystallographic data for this paper.

### 2.2. Polymerization Experiments

The ethylene polymerization and ethylene/α-olefine copolymerization techniques are described in detail in [21].

### 2.3. Polymer Characterization Methods

DSC was performed by a differential scanning calorimeter DSC-822e (Mettler-Toledo, Switzerland) at a heating rate of 10 °C/min in argon.

Viscosity average molecular weight of synthesized UHMWPE samples was calculated with the Mark–Houwink equation [35].

The technique for manufacturing-oriented films from UHMWPE nascent reactor powder and determining their mechanical characteristics is described in detail in [36].

Scanning electron microscopy investigations of morphologies of nascent reactor powders were carried out with a high-resolution Tescan VEGA3 SEM operated at 5 kV. As-polymerized particles were carefully deposited on SEM stubs, and the samples were coated with gold by a sputtering technique.

^13^C NMR spectra of ethylene/octene-1 copolymers (~5 wt % solutions in dichlorobenzene) were recorded at 150 °C on a Bruker Avance-400 spectrometer at 101 MHz.

Gel permeation chromatographic (GPC) analysis of copolymers was carried out at 135 °C with a Waters GPCV-2000 chromatograph equipped with two columns (PLgel, 5 μ and Mixed-C, 3007.5 mm) and a refractometer. 1,2,4-Trichlorobenzene was used as a solvent; the elution rate was 1 mL min^−1^. Molecular weights of polymers were determined using the universal calibration dependence relative to polystyrene standards with a narrow MW distribution: for polystyrene K = 2.88 × 10^−4^, α = 0.64; for PE, and K = 6.14 × 10^−4^, α = 0.67.

## 3. Results and Discussion

Commercially available (±) camphor was used as the initial compound for the synthesis of this group of ligands, the oxidation of which yielded camphoquinone (Figure 1). Further, the reduction of camphoquinone with sodium borohydride or its interaction with methyl magnesium iodide yielded ligands **L1-L2** (Figure 1) differing in the steric load of hydroxyl groups.

Alkoxo–titanium(IV) complexes **1–2** were obtained by the reaction of the ligands **L1–L2** with Ti(OiPr)_4_ in a toluene solution. All compounds were isolated in 69–82% yields as air-sensitive powders, which are soluble in aromatic hydrocarbons. Titanium–dichloride complexes **3–4** were synthesized by direct interaction of the ligands **L1–L2** with one equivalent of TiCl_2_(OiPr)_2_ in toluene. The compositions and structures of complexes **1–4** were confirmed by elemental analysis and ^1^H and ^13^C NMR spectroscopies. The integration of the NMR signals confirmed the presence of two isopropoxy groups per ligand unit in the reaction product. The structures of the complexes **1–2** were unambiguously established by X-ray diffraction study and are shown in Figure 1 along with the atomic numbering scheme. Experimental data for the X-ray diffraction studies of compounds and selected bond lengths and angles are given in Appendix A.

The complexes **1** and **2** crystallize in the triclinic space group P-1 with a half of the complex species being symmetry-independent; the appropriate symmetry element, the inversion center, is located in the geometric center of a Ti_2_O_2_ cycle. Each titanium(IV) ion coordinates two isopropoxy groups and two camphorquinone ligands that act both as a bridging ligand and a chelate ligand (Table 1). The resulting coordination polyhedron is a distorted square pyramid, as gauged by continuous symmetry measurements [37]. They measure how close the shape of the polyhedron is to a reference shape, such as an ideal square pyramid (**SPY-5**). The lower the value of an appropriate symmetry measurement, the better the fit is to a chosen polyhedron (Appendix A).

The catalytic activity of new titanium–diolate complexes **1–4** was studied in ethylene polymerization (Table 1). To activate precatalysts, binary activators {Et_2_AlCl or Et_3_Al_2_Cl_3_+Bu_2_Mg} at a molar ratio of Al/Mg = 3/1 [28,29], were used. For the activator {EtAlCl_2_ +Bu_2_Mg}, a molar ratio of Al/Mg = 2/1 was used since according to [21], it is precisely this ratio of the activator components that makes it possible to achieve the maximum productivity of diolate–titanium complexes.

For complex **1**, the effect of the nature of the organoaluminum compound (EtAlCl_2_, Et_2_AlCl, and Et_3_Al_2_Cl_3_) contained in the Al/Mg activator on the productivity of catalytic systems and the properties of the resulting polymer were studied (entries 1, 2, and 4; Table 1). The maximum activity was shown by the system containing EtAlCl_2_—a compound exhibiting the maximum Lewis acidity.

For the dichloride complex **3**, this pattern changes slightly: in a row Et2AlCl, Et3Al2Cl3 and EtAlCl2 there is a consistent increase in both the productivity of the system and the molecular weight of polymers (entries 15–17, Table 1).

Comparing the effect of the nature of the organoaluminum component of the activator on the properties of the resulting polymer, it can be noted that for dichloride complexes **3–4**, the replacement of Et_2_AlCl by Et_3_Al_2_Cl_3_ led to a very significant increase in molecular weight (by 2.3–2.6 times (entries 15 vs. 16 and 18 vs. 19, Table 1). In the case of alkoxide complexes **1–2**, this effect also manifested itself, but to a much lesser extent (no more than 1.2 times, entries 1 vs. 2 and 10 vs. 11); a similar trend was seen previously [21,26].

Preactivation (holding the precatalyst with a small amount of activator in a Schlenk tube for 24 h) led to a significant drop in activity, from 2629 to 1600 kg/mol·h·atm, with a simultaneous significant increase in the molecular weight of the polymer (from 5.9 up to 7.7 × 10^6^ Da). The replacement of the aromatic solvent toluene with the aliphatic one nefras was accompanied by a very significant decrease in activity (from 2629 to 460 kg/mol·h·atm and an equally noticeable increase in M_v_ from 5.9 to 8.5 × 10^6^, and the use of preactivation technique in aliphatic solvent made it possible to increase this value to 10.1 × 10^6^ Da (entries 5 and 6, Table 1).

For the alkoxo complexes **1** and **2**, the influence of the polymerization temperature on the activity and on the molecular weights of the resulting polyethylene was studied (Figure 2). It was established that both complexes exhibited a sufficiently high thermal stability: for complex **1** with an increase in the polymerization temperature from 10 to 50 °C, the activity increased by 20%. At a temperature of 70 °C, the activity remained quite high at 2300 kg/mol_Ti_·atm·h.

Complex **2** with the sterically more hindered ligands behaved somewhat differently: the maximum activity (3100 kg/mol_Ti_·atm·h.) appeared at 10 °C, and with an increase in the polymerization temperature, the activity consistently decreased to 1500 kg/mol_Ti_·atm·h. at 70 °C.

With an increase in the polymerization temperature, the processes of polymer chain termination were accelerated, which is reflected in a significant decrease in the molecular weights of the polymer. However, the polymers obtained at 50 °C and even 70 °C (for complex **2**) were ultra-high molecular weight polyethylenes. Thus, the process temperature allowed us to control the molecular weight of the resulting polymers.

The morphology and molecular weight are important characteristics of UHMWPE nascent reactor powder, which determine the efficiency of its processing into high modulus- and high strength-oriented materials. To examine the morphologies of these powders, scanning electron microscope (SEM) observations were performed (Figure 3). At a low magnification (Figure 3, top), the UHMWPE powder particles do not have a spherical shape, typical for the polymer obtained on classical Ti/Mg catalysts. The irregular shape and porous structure of the powder particles determine the low bulk density (0.05–0.088 g/cm^3^) of the obtained samples.

At high magnifications (Figure 3, bottom), the UHMWPE powder particles have a broccoli-like shape and differ in the number of fibrils connecting the globules.

The processing of the obtained UHMWPE reactor powders into high modulus-oriented films was carried out by preparing monolithic samples under pressure and shear deformation at an elevated temperature below the polymer melting point with a subsequent uniaxial drawing [36]. Mechanical tests were carried out for the samples oriented to the prefracture state, which varied for different samples (Table 2, Figure 4).

The UHMWPE nascent reactor powders obtained on bis-isopropoxo-titanium precatalysts **1** and **2** (entries 1,2,10, and 11) were processed into oriented films with approximately the same mechanical characteristics (Figure 4a). The nature of the organoaluminum compound (OAC) included in the Al/Mg activator did not significantly affect these parameters. The results obtained slightly exceeded those previously published for titanium complexes with diol ligands [17,18,21,25,26,27]; however, the reason may be not only the structure of the precatalysts, but also the polymerization temperature (in the cited works, polymerization was carried out at 30 °C). For comparison, the modulus value for commercially available gel-spun UHMWPE fiber, produced by the gel-spinning process, is 113 GPa [38]. The replacement of toluene with an aliphatic solvent nefras was reflected in the morphology of the UHMWPE reactor powder, namely, in an increase in the number of fibrils (Figure 3b,c), while the degree of crystallinity of these two samples of UHMWPE was determined by DSC and was the same at 55%. The presence of fibrillated elements prevented a uniform distribution of stress in the sample during orientation drawing and, as a result, led to a deterioration in the strength characteristics of film tapes (entries 2 and 3, Table 2). For oriented films from polymers obtained on titanium dichloride complexes **3** and **4**, the maximum values of the average tensile modulus were recorded when using an activator with Et_3_Al_2_Cl_3_.

An important condition for obtaining disentangled UHMWPE is to carry out the polymerization process at low temperatures, which allow to control the rates of polymer chain growth and its crystallization [39]. The fact that many samples obtained at elevated temperatures nevertheless turned out to be suitable for solid-phase processing (Figure 4b, curves 7 and 8) seems very promising to us.

The productivity of systems **1** and **2**/Et_3_Al_2_Cl_3_+Bu_2_Mg in the ethylene /1-octene copolymerization was noticeably lower than for the homopolymerization of ethylene; i.e., in this case, no positive effect of the comonomer was observed. The molecular weights of the copolymers (1.1–8.9 × 10^5^ Da) were also significantly lower than for the polyethylene samples (4.8 × 10^5^–1.01 × 10^7^ Da) (we can compare these data only at a qualitative level, since different methods of their determination were used).

The percentage of comonomer incorporation was low (1.4–4.6 mol% for precatalyst **1**), and it was obvious that an increase in the steric load at the metal center made it difficult for the bulk comonomer, 1-octene, to approach the reaction center. For complex **2**, this trend was more pronounced.

The ethylene/1-octene copolymerization process even more clearly demonstrated the increased thermal stability of complexes **1–2**: with an increase in the polymerization temperature from 10 to 50 °C, a noticeable increase in productivity was observed, which remains quite acceptable even at a temperature of 70 °C (Figure 5, Table 3). To our surprise, with increasing temperature, the molecular weight of the copolymers increased significantly, reaching a maximum at 50 °C.

## 4. Conclusions

In summary, new titanium(IV) complexes with OO^2−^-type diolate ligands in the presence of a binary cocatalysts {3Et_2_AlCl + Bu_2_Mg} or {1.5Et_3_Al_2_Cl_3_ + Bu_2_Mg} exhibited moderate to high activities toward ethylene polymerization (460–3260 kg/mol·h·atm). The M_v_ of the obtained polymer samples reached 10 million Da.

Compared to previously obtained titanium complexes with flexible aliphatic diolate ligands [15,16,17,18,19,20,21,22], complexes **1–4** with a rigid camphane framework were characterized by increased thermal stability. Complex **2** with an increased steric load at hydroxyl groups was able to produce UHMWPE even at a temperature of 70 °C. This UHMWPE sample was processed into an oriented film with a tensile strength of 1.6–2.0 GPa and an average tensile modulus of 108–123 GPa. Films obtained on the same precatalysts at a temperature of 10 °C were characterized by higher values of breaking strength up to 2.7 GPa and modulus up to 151 GPa.

Thus, directed changes in the ligand structure, namely the use of a rigid framework and an increase in the steric load of hydroxyl groups, seem to be a promising direction in the development of thermally stable precatalysts for the polymerization of olefins.

## Data Availability

The data presented in this study are available on request from the corresponding author.

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
