# Peer review of "New Titanium(IV)-Alkoxide Complexes Bearing Bidentate OO Ligand with the Camphyl Linker as Catalysts for High-Temperature Ethylene Polymerization and Ethylene/1-Octene Copolymerization"

_polymers, 2022, doi:10.3390/polym14214735_

Round 1
Reviewer 1 Report
The manuscript by Tuskaev et al. described preparation of four new titanium complexes bearing diolate ligands based on camphor moiety. The work is focused on utilization of the complexes (after activation with organoaluminiummagnesium activators) in homopolymerization of ethylene to UHMWPE and copolymerization of ethylene with 1-octene. The work is extension of authors previous investigation (ca 13 publication considering structures I-X in Chart 1). btw I am wondering whether there in no other group (beside manuscript authors) involved in the research of titanium diolate complexes. The complexes achieved high activity at industrially interesting temperatures above 50 degC, whereas maintained the formed polymer in UHMW territory (eg. complex 2 bearing sterically more hindered ligand). I see this as the most important result. The manuscript is not well written and need some corrections (for examples see below). However, I am supporting publication of the manuscript in Polymer after minor revisions metioned below:
- Scheme 1 hydroxyl groups are missing in ligands L1 and L2, also a hydrolysis should be marked in preparation of L2
- in experimental part NMR spectra are horribly interpreted. Eg. complex 1 page 2/line 85 signal at 0.86 ppm atributed to 12H (obviously CHMe2) could not be singlet. Please check again all spectra. In addition 13C NMR spectra are not given for complexes 2-4 (while it was measured at least for complex 2 - see SI)
- explain what does mean nefras
typos:
page4/line132 spectrometer at 10.616 Mz - suggesting 106.13??
Author Response
October 11, 2022
Manuscript Number: polymers-1927220
Dear Dr. Lisbeth Wang,
Thank you for your kind considering of our manuscript!
We thank Reviewers for providing constructive comments and help in improving the content of this paper. According to their comments, we made the following revisions (all changes in the text are highlighted in red):
Reviewer #1:
«I am wondering whether there in no other group (beside manuscript authors) involved in the research of titanium diolate complexes.»
We are also very surprised by the lack of publications of other scientific groups devoted to affordable, inexpensive and fairly effective precatalysts for the polymerization of olefins - titanium diolate complexes. Taking into account this fair comment of the reviewer, we have added a number of old works by other authors to the Introduction section: “Titanium alkoxide complexes are, perhaps, one of the most accessible and inexpensive precatalysts. It is known that titanium alkoxides in the presence of organoaluminum compounds are capable of catalyzing the polymerization of conjugated dienes [5-7] as well as the oligomerization of ethylene. The best known example is the Alphabutol process, wherein the combination Ti(OR)4–AlEt3 is employed for highly selective dimerisation of ethylene to 1-butene [8].
At the same time, simple titanium alkoxides as well as more sophisticated alkoxo-titanium complexes, have rarely been used for the polymerization of olefins. Thus, dichlorotitanium alkoxide complex 1 [(HOEt)Ti(m-OEt)OEt(Cl)2]2 (Chart 1), activated with MAO, catalyzes ethylene polymerization with activity up to 750 kg/mol h and propylene (up to 87 kg/mol h). This precatalyst displays unexpectedly a single site behavior [9]. The alkoxo-complexes 2-3 activated with MAO or AlEt3 catalyze the formation of polyethylene along with a percentage of oligomers. The bischelated complex 3 appears to be the most active [10]. A titanium(IV) dimeric complex 4 stabilized by a benzoin derivative in the presence of MAO catalyzes ethylene polymerization with an activity up to 300 kg /mol·atm·h., as well as the copolymerization of ethylene with norbornene The catalytic systems are characterized by long lifetime and the ability to produce high-molecular weight linear PE and vinyl-type PNB [11]. One of the few examples of homoleptic and heteroleptic Ti(IV) alkoxo-complexes 4-5 capable of polymerizing ethylene without the formation of oligomeric products is given in [12]. In the presence of Et3Al2Cl3, these compounds catalyzed the formation of low molecular weight PE.
As was shown by Y. V. Kissin et al., the addition of the organomagnesium compound, e.g. Bu2Mg to the mixture {Ti(OiPr)4-Et2AlCl} results in fairly active, cheap and affordable catalytic systems suitable for the polymerization of propylene [13] and the copolymerization of ethylene with higher olefins [14].
Chart 1. Examples of Ti(IV) complexes with diolate ligands used as olefin polymerization catalysts
Scheme 1 hydroxyl groups are missing in ligands L1 and L2, also a hydrolysis should be marked in preparation of L2
Necessary corrections have been made to scheme 1
- in experimental part NMR spectra are horribly interpreted. Eg. complex 1 page 2/line 85 signal at 0.86 ppm atributed to 12H (obviously CHMe2) could not be singlet. Please check again all spectra. In addition 13C NMR spectra are not given for complexes 2-4 (while it was measured at least for complex 2 - see SI)
We carefully reviewed the NMR spectra and made the necessary corrections.
explain what does mean nefras
Nefras - the petroleum fraction consisting of aliphatic hydrocarbons and boiling in a wider range 50–170°C and commonly used as an industrial solvent.
typos: page4/line132 spectrometer at 10.616 Mz - suggesting 106.13??
Bug fixed; frequency - 101 MHz
On behalf of all authors,
Prof. Boris M. Bulychev
Moscow State University,
119991 Moscow
Russia
Fax: +7(495)9393316
E-mail: bmbulychev@gmail.com,
Reviewer 2 Report
A long series of the syntheses of ultra-high molecular weight polyethylene, UHMWPE, using various titanium (IV) alkoxide complexes as precatalysts (Refs. 5-17 in this paper) is here extended to two complexes with camphor derived ligands. These newly modified catalysts allow to obtain a high yield of UHMWPs having very high molecular weight. There is no data on molecular weight distribution of these UHMWP which probably is high. These polymers could be processed to oriented films of high strength. The newly modified catalysts are distinguished from most others by a higher temperature range of catalytic activity. This research includes also studies on the synthesis of ethylene/octene-1 copolymers using these newly modified catalysts but their effectiveness in the incorporation of octene-1 to the copolymer was low.
The research was correctly performed and was well described, although the methodology and the manuscript organization were similar to those in previous papers of the Refs. 5-17 series. Results obtained in this research are important taking into account that the interest in the synthesis of UHMWPE is still high and the paper gives knowledge on the behavior of these newly modified catalysts when they contain various cocatalysts. This paper may be published in Polymer actually as it stands, but after the introduction of small corrections mentioned below. I do not understand why this paper is directed to the issue “Coordination Polymers”. The subject of this paper is not in line with the title of this issue.
In Figure 3 letters A B C should be marked in the inscription under the figure. Chart 1 should be completed by references where the structures showed in the chart were used. References 6 and 16 should be completed.
Author Response
October 11, 2022
Manuscript Number: polymers-1927220
Dear Dr. Lisbeth Wang,
Thank you for your kind considering of our manuscript!
We thank Reviewers for providing constructive comments and help in improving the content of this paper. According to their comments, we made the following revisions (all changes in the text are highlighted in red):
Reviewer #2:
There is no data on molecular weight distribution of these UHMWP which probably is high.
Indeed, the use of Al/Mg activators generally results in polymers with a broad molecular weight distribution. In this work, we cannot measure this parameter, since UHMWPE samples are insoluble in trichlorobenzene, which does not allow using the GPC method. It was also not possible to use the method of studying the melt rheology, because due to the high molecular weight, it is not possible to fix the cross-point of the curves of the elastic modulus and the loss modulus.
I do not understand why this paper is directed to the issue “Coordination Polymers”. The subject of this paper is not in line with the title of this issue.
We received an invitation from the editors to take part in this thematic issue. Then, we sent the abstract of this article to the editorial office and received approval.
In Figure 3 letters A B C should be marked in the inscription under the figure. Chart 1 should be completed by references where the structures showed in the chart were used. References 6 and 16 should be completed.
Necessary corrections have been made to Figure 3.
Chart 1 have been completed by references.
References 6 and 16 (now – 16 and 26) have been completed.
On behalf of all authors,
Prof. Boris M. Bulychev
Moscow State University,
119991 Moscow
Russia
Fax: +7(495)9393316
E-mail: bmbulychev@gmail.com,
Reviewer 3 Report
I think this work is not worth to be published in this high-impact factor journal because of all the similarities with previously published research papers.
So, besides the quality of the work, it is not worth to be published, unless the author decides to enlarge the generality to several cyclic structures able then to demonstrate that, the complex structure can improve the stability of the catalyst at a certain temperature. otherwise, it is no novel at all!!! also because in the conclusions, the author points out that this could be a good start for improving the stability of the catalyst without specifying its real power.
More specific is:
1. the abstract should be rewritten because it doesn't specify the real focus of the work
2. in the work the 'new' catalyst should improve the thermal stability but it doesn't seem to work as the author sepecified
3. they are all equal. For improving this work the author should compare several cyclic structures to generalize the method and really demonstrate the power of these new structures and how the cyclic organic structure can improve the thermal stability of the complex.
Author Response
October 11, 2022
Manuscript Number: polymers-1927220
Dear Dr. Lisbeth Wang,
Thank you for your kind considering of our manuscript!
We thank Reviewers for providing constructive comments and help in improving the content of this paper. According to their comments, we made the following revisions (all changes in the text are highlighted in red):
Reviewer #3:
- the abstract should be rewritten because it doesn't specify the real focus of the work
We have made changes to the abstract.
- in the work the 'new' catalyst should improve the thermal stability but it doesn't seem to work as the author sepecified
In this work, changes in the ligand structure allowed us to increase the polymerization temperature from 30° to 70°C, while the activity remained rather high. Moreover, the network density of macromolecular entanglements in UHMWPE samples obtained at elevated temperatures turned out to be rather low. This made it possible to use the method of solid-phase formation of high-strength oriented films for their processing. This result seems to us the most interesting, since polymerization at low temperatures was considered an indispensable condition for the synthesis of disentangled UHMWPE. Titanium diolate complexes, obtained by us earlier, in the given temperature range (50 - 70 °C) underwent a very significant deactivation.
- they are all equal. For improving this work the author should compare several cyclic structures to generalize the method and really demonstrate the power of these new structures and how the cyclic organic structure can improve the thermal stability of the complex.
Of course, we tried to increase the size of the substituents in close proximity to the hydroxyl groups. However, under mild synthesis conditions, the yields of dialkylated products are extremely low, while under more severe conditions, difficult-to-separate mixtures of products are formed due to rearrangements of the camphan skeleton. However, our group hopes to continue this work.
On behalf of all authors,
Prof. Boris M. Bulychev
Moscow State University,
119991 Moscow
Russia
Fax: +7(495)9393316
E-mail: bmbulychev@gmail.com,
Reviewer 4 Report
This manuscript describes synthesis, characterization, and catalytic properties of titanium(IV) complexes with diolate ligand. The titanium(IV) complexes are fully characterized. The use of a rigid campho linker enhances thermal stability of Ti catalyst. Besides, UHMWPEs are also prepared. I would recommend the publication of this manuscript after revising the following issues/problems.
(1) References are not enough, including Ti complexes with [O,O] ligands. Some other groups work should be cited.
(2) In copolymerization of ethylene and octene, broad distributions are observed. are GPC curves bimodal or monodal? Please provide GPC curves. Besides, are there two active species in systems for copolymerization? If yes, please explain them.
(3) To our surprise, with increasing temperature, the molecular weight of the copolymers increases significantly, reaching a maximum at 50 °C. This may be due to an increase in the solubility of the polymer in toluene. This reason is inaccurate. Maybe, octene is helpful to suppress chain transfer reaction.
(4) In the text, unit of activity should use kg/mol·h·atm instead of tons/mol·h·atm. Please keep consistent.
Author Response
October 11, 2022
Manuscript Number: polymers-1927220
Dear Dr. Lisbeth Wang,
Thank you for your kind considering of our manuscript!
We thank Reviewers for providing constructive comments and help in improving the content of this paper. According to their comments, we made the following revisions (all changes in the text are highlighted in red):
Reviewer #4:
(1) References are not enough, including Ti complexes with [O,O] ligands. Some other groups work should be cited.
Section Introduction and Chart 1 have been revised, 10 new references have been added: “Titanium alkoxide complexes are, perhaps, one of the most accessible and inexpensive precatalysts. It is known that titanium alkoxides in the presence of organoaluminum compounds are capable of catalyzing the polymerization of conjugated dienes [5-7] as well as the oligomerization of ethylene. The best known example is the Alphabutol process, wherein the combination Ti(OR)4–AlEt3 is employed for highly selective dimerisation of ethylene to 1-butene [8].
At the same time, simple titanium alkoxides as well as more sophisticated alkoxo-titanium complexes, have rarely been used for the polymerization of olefins. Thus, dichlorotitanium alkoxide complex 1 [(HOEt)Ti(m-OEt)OEt(Cl)2]2 (Chart 1), activated by MAO, catalyzes the polymerization of ethylene with activity up to 750 kg/mol h and propylene (up to 87 kg/mol h). This precatalyst displays unexpectedly a single site behavior [9]. The alkoxo-complexes with chelating ligands 2-3 activated with MAO or AlEt3 catalyze the formation of polyethylene along with a percentage of oligomers. The bischelated complex 3 appears to be the most active [10]. A titanium(IV) dimeric complex 4 stabilized by a benzoin derivative in the presence of MAO catalyzes ethylene polymerization with an activity up to 300 kg /mol·atm·h., as well as the copolymerization of ethylene with norbornene The catalytic systems are characterized by long lifetime and the ability to produce high-molecular weight linear PE and vinyl-type PNB [11]. One of the few examples of homoleptic and heteroleptic Ti(IV) alkoxo-complexes 4-5 capable of polymerizing ethylene without the formation of oligomeric products is given in [12]. In the presence of Et3Al2Cl3, these compounds catalyzed the formation of low molecular weight PE.
As was shown by Y. V. Kissin et al., the addition of the organomagnesium compound, e.g. Bu2Mg to the mixture {Ti(OiPr)4-Et2AlCl} results in fairly active, cheap and affordable catalytic systems suitable for the polymerization of propylene [13] and the copolymerization of ethylene with higher olefins [14].
Chart 1. Examples of Ti(IV) complexes with diolate ligands used as olefin polymerization catalysts
(2) In copolymerization of ethylene and octene, broad distributions are observed. are GPC curves bimodal or monodal? Please provide GPC curves. Besides, are there two active species in systems for copolymerization? If yes, please explain them.
Indeed, the use of Al/Mg activators generally results in polymers with a broad molecular weight distribution. We have included the MWD curves of copolymers in Supporting Information (Figures S22-S29), however at present we do not have enough experimental material for a full-fledged detailed analysis. In some cases, there are hints of bimodality on the GPC curves, but we cannot pinpoint the exact cause. Rather, there can be several reasons for the appearance of heterogeneous catalytically active sites, and all of them can be realized simultaneously. In the presence of significant excesses of organoaluminum, Ti(IV) complexes are easily reduced to Ti(III); the rate of this process depends on the nature of the OAC, the ligand environment of the metal, and temperature. Differences in the nature of active sites may also be due to the nature of the counteranion stabilizing the cationic alkylated titanium complex. Thus, as a result of the interaction of the components of Al/Mg activators {MgR2+AlknAlCl3-n}, in addition to the expected magnesium chloride and trialkyl aluminum, minor products of ionic nature are formed, and the compositions of aluminum-containing anionic products (capable of stabilizing a hypothetical active cationic particle) can also differ significantly. However, since we do not have clear evidence of these versions, we would prefer not to focus on this.
(3) To our surprise, with increasing temperature, the molecular weight of the copolymers increases significantly, reaching a maximum at 50 °C. This may be due to an increase in the solubility of the polymer in toluene. This reason is inaccurate. Maybe, octene is helpful to suppress chain transfer reaction.
You are absolutely right; our group has quite often encountered a situation where the addition of small amounts of higher olefins stabilized the catalytic system. We are trying to find reliable and theoretically sound explanations, so we do not write about it yet.
We have eliminated this controversial phrase.
(4) In the text, unit of activity should use kg/mol·h·atm instead of tons/mol·h·atm. Please keep consistent.
Corrected throughout the text on kg/mol·h·atm
On behalf of all authors,
Prof. Boris M. Bulychev
Moscow State University,
119991 Moscow
Russia
Fax: +7(495)9393316
E-mail: bmbulychev@gmail.com,